# Gene Expression Profiling of Peripheral Blood Mononuclear Cells in Type 2 Diabetes: An Exploratory Study

**DOI:** 10.3390/medicina58121829

**Published:** 2022-12-12

**Authors:** Hana M. A. Fakhoury, Muhammad Affan Elahi, Saud Al Sarheed, Mohammed Al Dubayee, Awad Alshahrani, Mahmoud Zhra, Arwa Almassri, Ahmad Aljada

**Affiliations:** 1Department of Biochemistry and Molecular Medicine, College of Medicine, Alfaisal University, Riyadh 11533, Saudi Arabia; 2College of Medicine, King Saud bin Abdulaziz University for Health Sciences, Riyadh 11481, Saudi Arabia; 3King Abdullah International Medical Research Center (KAIMRC), Riyadh 11481, Saudi Arabia; 4Department of Medicine, Ministry of National Guard Health Affairs (MNG-HA), Riyadh 11426, Saudi Arabia

**Keywords:** type 2 diabetes, metformin, metabolically activated macrophages, PBMC, qRT-PCR

## Abstract

*Background and Objectives*: Visceral obesity is associated with chronic low-grade inflammation that predisposes to metabolic syndrome. Indeed, infiltration of adipose tissue with immune–inflammatory cells, including ‘classical’ inflammatory M1 and anti-inflammatory ‘alternative’ M2 macrophages, causes the release of a variety of bioactive molecules, resulting in the metabolic complications of obesity. This study examined the relative expression of macrophage phenotypic surface markers, cholesterol efflux proteins, scavenger receptors, and adenosine receptors in human circulating peripheral blood mononuclear cells (PBMCs), isolated from patients with type 2 diabetes mellitus (T2DM), with the aim to phenotypically characterize and identify biomarkers for these ill-defined cells. *Materials and Methodology*: PBMCs were isolated from four groups of adults: Normal-weight non-diabetic, obese non-diabetic, newly diagnosed with T2DM, and T2DM on metformin. The mRNA expression levels of macrophage phenotypic surface markers (interleukin-12 (IL-12), C-X-C motif chemokine ligand 10 (CXCL10), C-C motif chemokine ligand 17 (CCL17), and C-C motif receptor 7 (CCR7)), cholesterol efflux proteins (ATP-binding cassette transporter-1 (ABCA1), ATP binding cassette subfamily G member 1 (ABCG1), and sterol 27-hydroxylase (CYP27A)), scavenger receptors (scavenger receptor-A (SR-A), C-X-C motif ligand 16 (CXCL16), and lectin-like oxidized LDL receptor-1 (LOX-1)), and adenosine receptors (adenosine A2A receptor (A2AR) and adenosine A3 receptor (A3R)) were measured using qRT-PCR. *Results*: In PBMCs from T2DM patients, the expression of IL-12, CCR7, ABCA1, and SR-A1 was increased, whereas the expression of CXCL10, CCL17, ABCG1,27-hydroxylase, LOX-1, A2AR and A3R was decreased. On the other hand, treatment with the antidiabetic drug, metformin, reduced the expression of IL-12 and increased the expression of 27-hydroxylase, LOX-1, CXCL16 and A2AR. *Conclusions*: PBMCs in the circulation of patients with T2DM express phenotypic markers that are different from those typically present in adipose tissue M1 and M2 macrophages and could be representative of metabolically activated macrophages (MMe)-like cells. Our findings suggest that metformin alters phenotypic markers of MMe-like cells in circulation.

## 1. Introduction

The pathological activation of the innate immune system in metabolic syndrome is well-established. Inflammation plays a key role in the development and progression of the multifaceted metabolic disorder type 2 diabetes mellitus (T2DM), which is characterized by peripheral insulin resistance and systemic glucolipotoxicity [1,2]. Indeed, inflammation may contribute to obesity-linked metabolic dysfunctions and is associated with polarization of the immune cells in various tissues such as adipose, skeletal, and liver tissues. Moreover, inflammation seems to play a role in the development of cardiovascular, renal, and ophthalmological complications.

Early in the disease course, immune–inflammatory cells, particularly macrophages, constitute the main source of inflammation. Evidence supports the idea that macrophage-induced inflammation contributes to insulin resistance in patients with T2DM, both locally and systemically [3,4]. Adipose tissue macrophages (ATMs) are the main source of inflammatory cytokines in adipose tissue, resulting in the pro-inflammatory ATM phenotype. However, the role of ATMs in potentiating insulin resistance in humans remains unclear. ATMs adopt a *‘*metabolically activated*’ (*MMe*)* phenotype, which is distinct from M1 (pro-inflammatory)-like and M2 (anti-inflammatory)-like types [5]. Indeed, MMe represents a pro-inflammatory phenotype that produces large amounts of pro-inflammatory cytokines in response to metabolic conditions in human adipose macrophages [6]. The inability of classical models of macrophage inflammation to accurately represent the phenotype of ATMs in vivo represents a limiting factor in delineating ATMs role in insulin resistance and T2DM. Systemic changes in insulin resistance and T2DM affect macrophage metabolism, thus resulting in altered metabolism and distinct activation phenotypes in the adipose tissue and the periphery. It is noteworthy that we have previously described a distinctive phenotype of peripheral blood mononuclear cells (PBMCs) in T2DM which is different from the pattern typically present in M1 and M2-like cells [7].

PBMCs are a sentinel sample, consisting of monocytes and lymphocytes that are sensitive to a variety of stimuli such as ongoing disease processes [8]. In early-onset T2DM, the PBMCs’ response is active as well [9]. In addition, an increase in pro-inflammatory cytokines and transcription factors in PBMCs is associated with obesity and T2DM [10,11]. Blood monocytes release bioactive molecules, such as pro-inflammatory cytokines and lipoprotein phospholipase A_2_, which play a key role in the onset and development of the innate immune system [9,12]. This inflammatory state is induced by several pro-inflammatory transcription factors including nuclear factor kappa B (NF-κB), which is responsible for triggering transcription of several pro-inflammatory genes and has been shown to be increased in PBMCs from T2DM [10]. However, inflammation and stress cause hypoxia and accumulation of extracellular adenosine [13,14] resulting in the activation of the adenosine receptors, adenosine A2A receptor (A2AR) and adenosine A3 receptor (A3R), which inhibit NF-κB signaling pathways, enabling them to exert anti-inflammatory effects [15]. Importantly, the role of the adenosine receptors has not been properly evaluated in humans. Nevertheless, a relationship between the A3 receptor and cholesterol has been suggested [16], and adenosine receptors are currently investigated as therapeutic targets in clinical trials for the treatment of various diseases [17].

Scavenger receptors are key regulators of inflammatory diseases, such as atherosclerosis [18]. Moreover, hyperglycemia has been shown to increase the expression of scavenger receptors, namely scavenger receptor-A (SR-A), CD36, and lectin-like oxidized LDL receptor-1 (LOX-1) [19]. Interestingly, adipose tissue scavenger receptors have been demonstrated to be strongly associated with insulin resistance [20]. Although circulating monocytes are known to express little or no scavenger receptors, their mRNA and protein expression was shown to be increased following monocytes’ differentiation into macrophages in culture [21]. On the other hand, cholesterol efflux pathways exert an anti-inflammatory effect by suppressing inflammation and inflammasome activation in macrophages [22].

In this study, we examined the expression levels of many phenotypic markers by PBMCs in T2DM to explore the altered metabolism and distinct activation phenotypes of PBMCs in vivo, which may assist in the characterization of an MMe-distinctive phenotype. The expression level of scavenger receptors (SR-A1, C-X-C motif ligand 16 (CXCL16), and LOX-1), cholesterol efflux proteins (ATP-binding cassette subfamily G member 1 (ABCG1), ATP-binding cassette transporter-1 (ABCA1), and sterol 27-hydroxylase (CYP27A)), as well as macrophage phenotype-specific surface markers (interleukin-12 (IL-12), C-X-C motif chemokine ligand 10 (CXCL10), C-C motif chemokine ligand 17 (CCL17), and C-C motif receptor 7 (CCR7)) were measured in PBMCs isolated from four groups of adults: normal-weight non-diabetic, obese non-diabetic, newly diagnosed with T2DM, and T2DM on metformin. Stimuli associated with T2DM promote PBMC-distinct mechanisms and surface markers activation in circulation. Collectively, the data from this study provide important mechanistic insights into pathways that drive the metabolic-disease-specific phenotype of PBMCs in circulation.

## 2. Materials and Methods

### 2.1. Subjects

The study, approved by King Abdullah International Medical Research Center (KAIMRC) Institutional Review Board (reference number 101/15; date of approval 17 February 2015), was conducted on four groups of adults: 30 normal-weight non-diabetic (BMI: 23.0 ± 0.3 kg/m^2^), 30 obese non-diabetic (BMI: 39.1 ± 1.7 kg/m^2^), 20 newly diagnosed with T2DM (BMI: 32.5 ± 1.9 kg/m^2^), and 30 patients with T2DM on metformin (BMI: 40.5 ± 1.5 kg/m^2^; 9 months-15 years duration). While most patients were on daily doses of metformin ranging between 1000 and 2000 mg, 5 patients had a smaller dose of 100–500 mg, and only 3 patients had a higher dose of 3000–4500 mg. In addition, 5 patients were on combined therapy of metformin and insulin. Few subjects from the obese, obese T2DM, and obese T2DM on metformin groups were on stable doses—for the last two months of their participation in the study—of statins or other cholesterol-lowering agents, angiotensin-converting enzyme inhibitors (ACE-I) or other anti-hypertensives, NSAIDS or antioxidants. All subjects were enrolled into the study following medical screening at King Abdulaziz Medical City in Riyadh, Saudi Arabia. Anthropometric measurements were measured (blood pressure, weight, height, waist, and hip circumferences along with complete blood count *(*CBC*),* blood urea nitrogen (BUN), creatinine, fasting blood sugar, liver function tests, and lipid profile) following 12 h of fasting before their venipuncture appointment. Exclusion criteria included coronary event or procedure in the previous 3 months, liver disease, renal impairment, history of drug or alcohol abuse, and steroid therapy.

### 2.2. Measurement of Plasma Insulin, LDL, HDL, Triglycerides, HbA1c and HOMA-IR

Levels of plasma insulin*,* low-density lipoprotein (LDL), high-density lipoprotein (HDL)*,* triglycerides, and hemoglobin A1c (HbA1c) were measured on fasting samples using fully automated methods at the central clinical laboratories at the Ministry of National Guard Health Affairs (MNGHA). Insulin resistance score (HOMA-IR) was calculated using the following formula: HOMA-IR = (insulin (mU/L) × glucose (mmol/L)*)*/22.5 [23].

### 2.3. Isolation of PBMCs

A total of 10 mL of blood was obtained from all participants and collected in EDTA-containing tubes. Blood samples were mixed with 10 mL of PBS and layered over 15 mL of Ficoll–Hypaque (50 mL Leucosep Tubes, Greiner Bio-One North America Inc, Monroe, NC, USA). Samples were centrifuged at room temperature for 30 min at 450× *g*. The PBMC layer was harvested with a pipette then washed with PBS. An aliquot of 50 µL Qiagen RNALater was added, and the samples were stored at −80 °C (Figure 1).

### 2.4. Quantitative Real-Time Polymerase Chain Reaction Analysis

Total RNA was isolated using Ambion Aqueous kit (Ambion). Total RNA was treated with DNase I to remove any genomic DNA contaminants. RNA electrophoresis was run on an Agilent Bioanalyzer 2100 to check for the quantity and quality of isolated RNA. cDNA was synthesized from 1 µg of RNA using a first strand cDNA synthesis Kit (Millipore, USA) followed by quantitative real-time PCR (qRT-PCR) to detect and quantitate IL-12, CXCL10, CCL17, CCR7, ABCA1, ABCG1, CYP27A, SR-A1, LOX-1, CXCL16, A2aR, A3R, Ubiquitin C, Ribosomal Protein L13 (RPL13), and Cyclophilin A. qRT-PCR amplification using primers purchased from Bio-Basic Canada Inc. (Markham, ON, Canada) was performed on a 7900HT Fast Real-Time PCR System (Applied Biosystems, San Francisco, CA, USA). qRT-PCR reactions (2 µL of cDNA, 10 µL of 2×SYBR^®^ Green Master mix (150 mM Tris, pH 9.2, 40 mM (NH_4_)_2_SO_4_, 5 mM MgCl_2_, 0.02% Tween-20, 0.4 mM dNTPs, 1.25 Units *Taq* Polymerase, and 1× SYBR Green) and 0.5 µL of 20 μM gene-specific primers (Table 1)). A melt curve at the end of PCR cycling was performed. Normalization to several housekeeping genes (Cyclophilin A, RPL13, and Ubiquitin C) showed similar trends. mRNA expression levels were normalized to Cyclophilin A using the 2^−ΔΔ*CT*^ method for qRT-PCR relative quantitation [24].

### 2.5. Statistical Analysis

SigmaStat software ver. 3.0 (Jandel Scientific, San Rafael, CA, USA) was used for statistical analysis. Changes in mRNA expression levels were computed for qRT-PCR results, and analysis was conducted using one-way ANOVA on ranks for all analytes measured in this study as the Kolmogorov–Smirnov test normality distribution test failed. Dunn’s test for pairwise comparisons and comparisons against the normal-weight group was used. Correlation analysis between age and expression of macrophage phenotypic markers, cholesterol efflux proteins, scavenger receptors, or adenosine receptors was performed using Spearman’s rank correlation. *p*-value < 0.05 was used to assess significance for all statistical analyses. Results are presented as mean ± standard error of the mean (SEM).

## 3. Results

### 3.1. Demographic Data of Study Participants

The demographic data of the study participants (Table 2) indicate that groups 3 and 4 were significantly older than the normal-weight and obese groups. However, there was no significant correlation (Pearson *r* correlation) between age and the expression levels of macrophage phenotypic markers, cholesterol efflux proteins, scavenger receptors, or adenosine receptors examined in the study. To examine the effect of gender in the 12 analyzed biomarkers (IL-12, CXCL10, CCL17, CCR7, ABCA1, ABCG1, 27-hydroxylase, SR-A1, CXCL16, LOX-1, A2AR, and A3R), a chi-square test was performed. Gender was statistically non-significantly associated with the different biomarkers tested (χ^2^(10) = 0.828, *p* = 1.000). However, gender was statistically significantly associated among different study groups (χ^2^(3) = 9.50, *p* = 0.023).

Despite being on medication, the T2DM on metformin group had significantly higher blood glucose and HbA1c compared to the normal-weight and obese groups. The T2DM group had significantly higher LDL when compared to the normal-weight disease-free group (*p <* 0.05). However, the T2DM on metformin group had significantly lower LDL when compared to the T2DM group, suggesting that metformin restores LDL values to normal.

### 3.2. Macrophage Phenotypic Markers’ Expression

IL-12 expression was significantly higher in the T2DM group (*p <* 0.05) when compared to the lean and obese groups. This increased expression was significantly lowered by metformin treatment (*p <* 0.05). Moreover, the expression of CCR7 was significantly higher in the T2DM group when compared to both non-diabetic groups (*p <* 0.05). Metformin had no significant effect on CCR7. On the other hand, CXCL10 and CCL17 mRNA expression was lower in the T2DM group when compared to both non-diabetic groups (*p <* 0.05). Metformin treatment conferred no significant change in the expression of CXCL10 and CCL17 (Figure 2).

### 3.3. Expression of Cholesterol Efflux Proteins

A significant increase in the expression of ABCA1 was observed in the T2DM group when compared to the non-diabetic groups (*p <* 0.05). This increase was not altered by metformin treatment. In contrast, ABCG1 expression was reduced in the T2DM group when compared to the disease-free groups (*p <* 0.05). Moreover, 27-hydroxylase expression was significantly lower in the T2DM group when compared to non-diabetic groups (*p <* 0.05). Metformin significantly increased the expression of sterol 27-hydroxylase in the PBMCs of the T2DM on metformin group compared to the T2DM group (*p <* 0.05, Figure 3).

### 3.4. Expression Analysis of Scavenger Receptors

SR-A1 expression was significantly higher in the T2DM group compared to the disease-free groups (*p <* 0.05). Although metformin treatment reduced the expression of SR-A1, the reduction in SR-A1 expression by metformin did not reach statistical significance (*p >* 0.05). Moreover, LOX-1 expression was not statistically significantly lower in the T2DM group compared to the disease-free groups (*p >* 0.05), but metformin administration significantly increased LOX-1 expression to levels higher than in the T2DM group (*p <* 0.05). This contradicts the reported observation that LOX-1 expression is increased in THP-1 cells induced by interferon-γ and LPS to M1 [25]. CXCL16 expression was not statistically significantly higher in the T2DM group compared to the disease-free groups (*p >* 0.05), and metformin treatment increased CXCL16 expression to a statistically significant level (*p <* 0.05, Figure 4).

### 3.5. Adenosine Receptors A2AR and A3R Expression

A significant decrease in A2A and A3 receptor expression was observed in the T2DM groups as opposed to the disease-free groups (*p <* 0.05). However, metformin treatment led to a considerable increase in the expression of A2AR (*p <* 0.05) but not in A3R expression ((*p >* 0.05, Figure 5).

### 3.6. Summary of the Changes in the Relative Expression of the Studied Biomarkers

The mRNA expression of all markers examined in this study is summarized in Figure 6. In PBMCs from T2DM patients, the expression of IL-12, CCR7, ABCA1, and SR-A1 was increased, whereas the expression of CXCL10, CCL17, ABCG1, 27-hydroxylase, A2AR and A3R was decreased. On the other hand, treatment with metformin, was associated with increased expression of CCR7, ABCA1, SR-A1, LOX-1, and CXCL16 and decreased expression of CXC10, CCL17, ABCG1, and A3R.

## 4. Discussion

The circulating PBMCs of T2DM have a different pattern of phenotypic markers than the patterns typically present in M1 and M2 macrophages that could be representative of MMe-like cells [7]. Higher expression levels of CD16, interleukin-6 (IL-6), inducible nitric oxide synthase *(*iNOS*),* tumor necrosis factor alpha or *(*TNFα), and CD36 have been reported in the PBMCs of T2DM which is consistent with the M1 macrophage-like phenotype. In addition, the PBMCs of T2DM have higher expression levels of mannose receptor (CD206), an M2-specific marker [7]. A majority of the data regarding macrophage polarization are from studies conducted on murine models despite significant differences between human and murine macrophages. Other macrophage surface markers (IL-12, CXCL10, CCL17, and CCR7) were analyzed in this study to further characterize human circulating PBMC phenotypes in T2DM and compare them to macrophage phenotypes since information on human macrophage polarization remains sparse. Several studies have attempted to characterize MMe in the adipose tissue of obese and diabetic subjects [26,27,28]. Ex vivo treatment of monocyte-derived macrophages with glucose, insulin, and palmitate induced a different phenotype than the M1 macrophage phenotype [6]. An in vitro study utilizing human monocytic cell lines classified IL-12, CCR7, and CXCL10 as M1 macrophage markers, and CCL17 as an M2 marker [25]. The increased expression of IL-12 and CCR7 in the PBMCs of T2DM is consistent with M1-like cells. Metformin treatment reduced IL-12 levels indicating that it reduces the M1 macrophage-like phenotype. CCL17, which is released from alternatively activated macrophages, serves to prevent a generation of classically activated macrophages, and is considered an M2 marker [26], was inhibited in the PBMCs of T2DM. Thus, our previously published CD163 results [7] and current CCL17 results are consistent with an anti-inflammatory role of CD163 and CCL17 (M2 markers) as they are reduced in T2DM [7]. Metformin treatment resulted in an increase in CD163 and CCL17, suggesting that it induces a phenotype similar to the M2-like phenotype [7]. However, the expression of CXCL10, also known as interferon gamma (IFN-γ)-inducible protein 10, was inhibited in the PBMCs of T2DM [7]. CXCL10, secreted by leukocytes and tissue cells, functions as a chemoattractant, mainly for lymphocytes. A recent study reported CXCL10 and CXCL11 as potential biomarkers for the onset of adipose tissue inflammation during obesity with CXCL11 expression correlation with NF-κB expression [27]. This discrepancy could stem from the heterogeneity of PBMCs and further studies are needed to examine the role of the CXCL subfamily of chemokines (CXCL9, CXCL10, and CXCL11; angiostatic chemokines) in adipose tissue inflammation and their utilization as biomarkers for the MMe phenotype.

Inflammation and stress cause hypoxia and an accumulation of extracellular adenosine [13,14]. Adenosine receptors’ (A2AR and A3R) activation inhibits NF-κB signaling pathways, enabling them to exert anti-inflammatory effects [15]. The NF-κB pathway is responsible for triggering the transcription of several pro-inflammatory genes. The NF-κB pathway plays a pivotal role in insulin resistance and ATM activation [28]. On the other hand, A2AR plays a protective role in obesity-associated adipose tissue inflammation by suppressing macrophage pro-inflammatory activation, including inhibition of the NF-κB pathway [29]. High-fat-diet (HFD) feeding of A2AR-disrupted mice increased adipose tissue inflammation and adipose tissue insulin resistance [29]. Similarly, the activation of A3R resulted in NF-κB pathway inhibition [30]. In our study, A2AR and A3R expression levels were significantly lower in the PBMCs from T2DM. These results suggest that activation of the NF-κB pathway, with the subsequent activation of inflammation, reported in T2DM could be mediated by A2AR and A3R inhibition. Thus, adenosine could be utilized as a new strategy to regulate metabolic homeostasis through the modulation of adipocyte–macrophage interaction. Interestingly, A2AR has been shown to diminish foam cell formation by increasing the expression and function of cholesterol 27-hydroxylase, an enzyme involved in the conversion of cholesterol to oxysterols [31] and enhancement of cholesterol efflux [32]. A2AR and cholesterol 27-hydroxylase expression are inhibited in this study in T2DM. Metformin increased the expression levels of A2AR and 27-hydroxylase beyond the levels expressed in T2DM, suggesting that it enhances cholesterol efflux from peripheral tissues by upregulating A2AR and 27-hydroxylase.

ABCA1 and ABCG1 play a key role in mediating the efflux of cholesterol from peripheral cells. ABCA1 and ABCG1 promote unidirectional cholesterol efflux to lipid-poor apolipoprotein A-I (apoA-I), apoE, or HDL particles [33]. The M1 phenotype has been shown to express high levels of the ABCA1 protein [25] and lower levels of SR-B1 involved in cholesterol efflux compared to M2 macrophages [34]. No changes in the expression of ABCG1 in M1 and M2 subsets were observed by Littlefield et al. [25] compared to Waldo et al. [34]. Additionally, ABCG1 expression in metabolic syndrome patients has been shown to be significantly lower [35]. ABCG1 has been reported to promote LPL-dependent triglyceride storage in adipocytes [36] and to modulate ATM cholesterol content in obesity and weight loss regimes, leading to an alteration in M1 to M2 ratio [37]. On the other hand, the ABCA1 transporter, which plays a key role in the first steps of the reverse-cholesterol-transport pathway by mediating lipid efflux from macrophages, stimulates the production of more monocytes, leading to an exacerbation of inflamed-tissue macrophages [38]. These observations could explain the increased levels of ABCA1 and reduced ABCG1 expression in the PBMCs of T2DM observed in this study.

Adipose tissue scavenger receptors (SR-A and LOX-1) are strongly associated with insulin resistance [20]. The PBMCs of T2DM displayed increased expression of the major scavenger receptor responsible for modified lipid uptake: SR-A1. Similarly, an increase in CD36 expression in T2DM has recently been demonstrated [39]. Increased CD36 expression in both macrophages and adipocytes of the adipose tissue induces inflammation in obesity [40]. CXCL16 can serve as an adhesion molecule for immune cells expressing CXCR6. It also acts as a scavenger receptor for oxLDL. Although CXCL16 and LOX-1 levels in the PBMCs from T2DM were not statistically significant, increased expression of CXCL16 in M1 macrophages [25] and increased expression of LOX-1 in M2-polarized macrophages have been observed which is consistent with the current model for increased lipid uptake in M2 macrophages [25,41]. However, CXCL16 and LOX-1 levels were significantly higher in T2DM on metformin. Several observations suggest that metformin exhibits anti-atherogenic properties and is associated with reduced cardiovascular morbidity and mortality in patients with diabetes [42,43,44,45,46]. A significant and substantial increase in CXCL16 level in T2DM on metformin demonstrated that this drug has an athero-protective role. This is in light of the finding that CXCL16−/−/LDLR−/− mice have accelerated the progression of atherosclerosis [47].

A significant shortcoming of this study is the heterogeneity of the cells utilized, and purified monocytes could be a better model for such studies. Nevertheless, monocyte purification could activate monocytes bringing about changes in the expression of pro-inflammatory mediators and phenotypic markers [48]. Moreover, we elected to utilize qRT-PCR to quantitate the phenotypic markers of ATM which is a common practice [49]. Contradictions concerning white blood cells’ subset phenotypes and function, as a result of discrepancies in reliable gating strategies for flow cytometric characterization, antibody specificities, and cell purification protocols, have been reported [50,51]. Moreover, the lack of quantitation of the studied phenotypic markers in this study at the protein level is another limitation of our study as mRNA expression levels may not correlate with the protein expression levels. However, the semi-quantitative Western blotting technique requires large quantities of PBMNCs; hence, the quantification of proteins using tandem mass spectrometry could be a better alternative for future studies. The correlation of the phenotypic markers with plasma cytokines represents another future study as well. Another shortcoming of the study is the statistically significant differences in the age of normal-weight subjects compared with the age of obese participants with T2DM and participants with T2DM on metformin. However, there was no significant statistical correlation between the mRNA expression of the phenotypic markers measured in this study and the subjects’ age.

## 5. Conclusions

Data from this exploratory study suggest the presence of a PBMC phenotype that shares a similarity between M1 and M2 phenotypes and could represent an MMe macrophage-like phenotype. Metformin modulates the phenotypic characteristics of PBMCs resulting from metabolic stress present in T2DM. Future studies will attempt to quantitate these phenotypic markers in isolated monocytes from T2DM at the mRNA and protein levels.

## Figures and Tables

**Figure 1 medicina-58-01829-f001:**
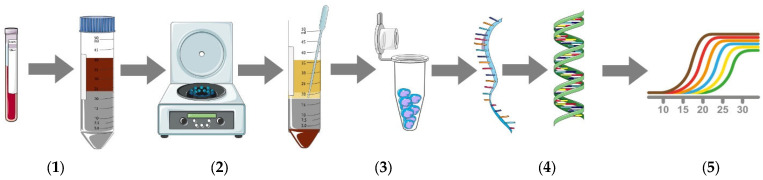
A schematic diagram for the study methods: (**1**) Blood Collection; (**2**) Mononuclear Cell Isolation (Ficoll–Hypaque); (**3**) RNA Isolation; (**4**) Reverse Transcription and cDNA synthesis; (**5**) qRT-PCR (Optimization, C_T_ determination, and Statistical analysis).

**Figure 2 medicina-58-01829-f002:**
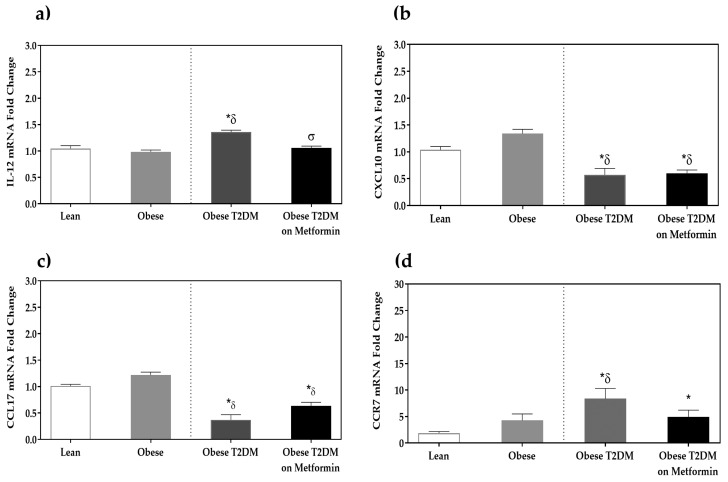
mRNA expression of macrophage markers in the peripheral blood mononuclear cells (PBMCs) of lean, obese, obese T2DM, and obese T2DM on Metformin groups: (**a**) IL-12; (**b**) CXCL10; (**c**) CCL17; and (**d**) CCR7. Results are presented as Mean ± SEM. One-way ANOVA on ranks, *p <* 0.001 for IL-12, CXCL10, CCL17, and CCR7, followed by Dunn’s test for pairwise comparisons and comparisons against normal-weight group, *******
*p <* 0.05 vs. normal-weight subjects; ***^δ^*** *p <* 0.05 vs. obese; ***^σ^***
*p <* 0.05 vs. T2DM subjects.

**Figure 3 medicina-58-01829-f003:**
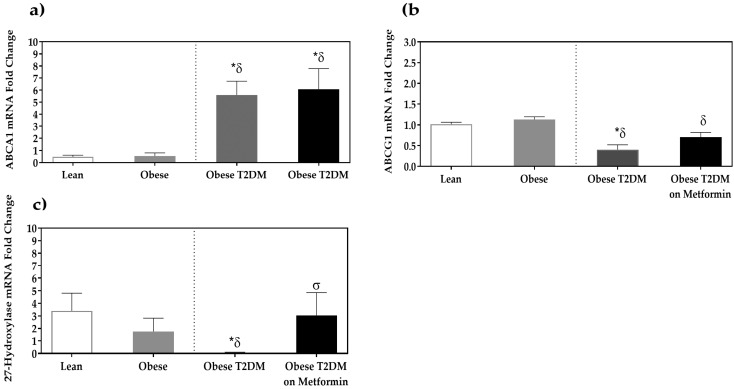
mRNA expression of cholesterol efflux proteins in the PBMCs of lean, obese, T2DM and T2DM on Metformin groups: (**a**) ABCA1; (**b**) ABCG1; and (**c**) Sterol 27-hydroxylase. Results are presented as Mean ± SEM. One-way ANOVA on ranks, *p <* 0.001 for ABCA1, ABCG1 and Sterol 27-hydroxylase, followed by Dunn’s test for pairwise comparisons and comparisons against normal-weight group, *******
*p <* 0.05 vs. normal-weight subjects; ***^δ^*** *p <* 0.05 vs. obese; ***^σ^***
*p <* 0.05 vs. T2DM subjects.

**Figure 4 medicina-58-01829-f004:**
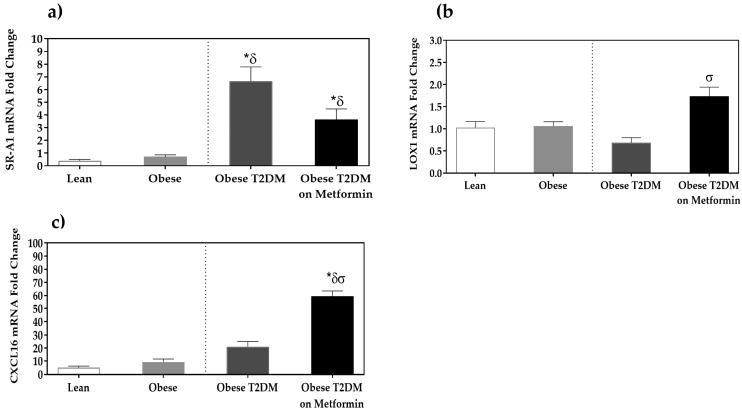
mRNA expression of scavenger receptors in the PBMCs of lean, obese, T2DM and T2DM on Metformin groups: (**a**) SR-A1; (**b**) LOX-1; and (**c**) CXCL16. Results are presented as Mean ± SEM. One-way ANOVA on ranks, *p <* 0.001 for SR-A1, LOX-1, and CXCL16, followed by Dunn’s test for pairwise comparisons and comparisons against normal-weight group, *******
*p <* 0.05 vs. normal-weight subjects; ***^δ^*** *p <* 0.05 vs. obese; ***^σ^***
*p <* 0.05 vs. T2DM subjects.

**Figure 5 medicina-58-01829-f005:**
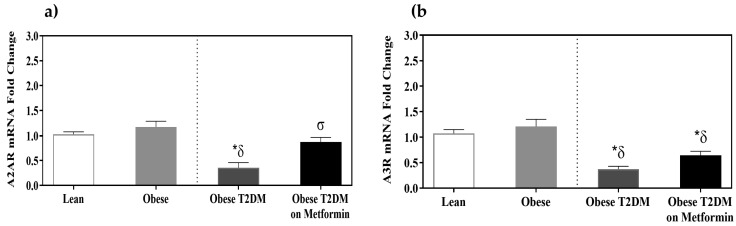
mRNA expression of adenosine receptors in the PBMCs of lean, obese, T2DM and T2DM on Metformin groups: (**a**) A2AR; and (**b**) A3R. Results are presented as Mean ± SEM. One-way ANOVA on ranks, *p <* 0.001 for A2AR, and A3R, followed by Dunn’s test for pairwise comparisons and comparisons against normal-weight group, *******
*p <* 0.05 vs. normal-weight subjects; ***^δ^*** *p <* 0.05 vs. obese; ***^σ^***
*p <* 0.05 vs. T2DM subjects.

**Figure 6 medicina-58-01829-f006:**
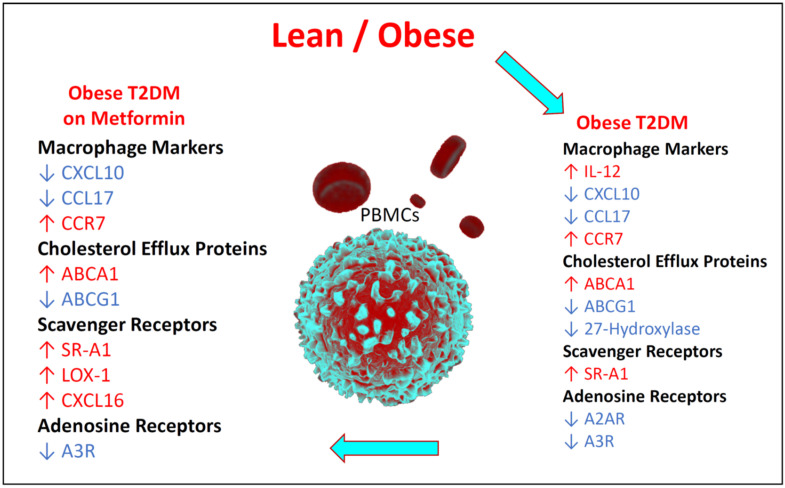
Summary of mRNA expression of all biomarkers examined in this study.

**Table 1 medicina-58-01829-t001:** List of specific primer sequences for all primers used in qRT-PCR. IL-12: interleukin-12, CXCL10: C-X-C motif chemokine ligand 10, CCL17: C-C motif chemokine ligand 17, CCR7: C-C motif receptor 7, ABCA1: ATP-binding cassette transporter-1, ABCG1: ATP binding cassette subfamily G member 1, CYP27A: sterol 27-hydroxylase, SR-A: scavenger receptor-A, CXCL16: C-X-C motif ligand 16, LOX-1: lectin-like oxidized LDL receptor-1, A2AR: adenosine A2A receptor, A3R: adenosine A3 receptor

Primer	Sense (5′→3′)	Anti-Sense (5′→3′)	Accession Number
IL-12 (IL12RB2)	AAAGGACATCTGCGAGGAAAGTTC	CGAGGTGAGGTGCGTTTATGC	NM_001559.2
CXCL10	GAAAGCAGTTAGCAAGGAAAGGTC	ATGTAGGGAAGTGATGGGAGAGG	NM_001565.4
CCL17	CGGGACTACCTGGGACCTC	CCTCACTGTGGCTCTTCTTCG	BC112068.1
CCR7	TGGTGGTGGCTCTCCTTGTC	TGTGGTGTTGTCTCCGATGTAATC	NM_001838.4
ABCA1	GAAGTACATCAGAACATGGGC	GATCAAAGCCATGGCTGTAG	NM_005502.4
ABCG1	CAGGAAGATTAGACACTGTGG	GAAAGGGGAATGGAGAGAAG	NM_016818.2
CYP27A	AAGCGATACCTGGATGGTTG	TGTTGGATGTCGTGTCCACT	NM_000784.4
SR-A1	CTCGTGTTTGCAGTTCTCA	CCATGTTGCTCATGTGTTCC	NM_138715.3
LOX-1	TTACTCTCCATGGTGGTGCC	AGCTTCTTCTGCTTGTTGCC	NM_002543.4
CXCL16	ACTACACGACGTTCCAGCTCC	CTTTGTCCGAGGACAGTGATC	NM_022059.4
A2AR	CGAATTCAACCTGCAGAACGTCACC	TCGAATTCGCGGTCAATGGCGATG	NM_001278497.1
A3R	ACCACTCACAGAAGAATATG	ACTTAGCCGTCTTGAACTCC	L22607.1
Ubiquitin C	ACTACAACATCCAGAAAGAGTCCA	CCAGTCAGGGTCTTCACGAAG	NM_021009.6
RPL13	AACAAGTTGAAGTACCTGGCTTTC	TGGTTTTGTGGGGCAGCATA	NM_000977.3
Cyclophilin A	CCCACCGTGTTCTTCGACAT	TTTCTGCTGTCTTTGGGACCTT	NM_021130.4

**Table 2 medicina-58-01829-t002:** Demographic data of the study participants.

Group	Gender	Age	BMI	Glucose	Insulin	HOMA-IR	LDL	HDL	Triglycerides	HbA1c
		(Years)	kg/m^2^	(mmol/L)	µIU/mL		(mmol/L)	(mmol/L)	(mmol/L)	(%)
Normal weight	N = 30, 18 Males, 12 Females	25.7 ± 1.1	23.0 ± 0.3	5.1 ± 0.1	4.8 ± 0.60	1.1 ± 0.1	2.54 ± 0.26	1.35 ± 0.04	0.83 ± 0.09	5.8 ± 0.10
Obese	N = 30, 11 Males, 19 Females	35.1 ± 2.3	39.1 ± 1.7 ***	5.4 ± 0.1	10.0 ± 1.54 ***	2.1 ± 0.3 ***	3.04 ± 0.16	1.18 ± 0.05	1.22 ± 0.12	5.6 ± 0.25
Obese with T2D	N = 20, 15 Males, 5 Females	48.4 ± 3.0 **^δ^*	32.5 ± 1.9 ***	10.0 ± 1.1 **^δ^*	7.4 ± 2.57 ***	2.8 ± 0.8 ***	3.53 ± 0.19 ***	1.02 ± 0.05 **^δ^*	1.97 ± 0.23 **^δ^*	8.0 ± 0.62 **^δ^*
T2DM on Metformin	N = 30, 12 Males, 18 Females	47.1 ± 2.0 **^δ^*	40.5 ± 1.5 ***	10.0 ± 0.8 **^δ^*	7.8 ± 1.52 ***	4.1 ± 1.3 ***	2.60 ± 0.17 ^σ^	0.99 ± 0.04 **^δ^*	1.56 ± 0.16 ***	8.7 ± 0.35 **^δ^*

Results are presented as Mean ± S.E.M; one-way ANOVA on ranks, *p <* 0.001 for all analytes, followed by Dunn’s test for pairwise comparisons and comparisons against normal-weight group, *******
*p <* 0.05 vs. normal-weight subjects; ***^δ^*** *p <* 0.05 vs. obese; ***^σ^***
*p <* 0.05 vs. type 2 diabetes mellitus (T2DM) subjects.

## Data Availability

The data that support the findings of this study are available from the corresponding author (AA), upon reasonable request.

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
