# Peer review of "Gene Expression Profiling of Peripheral Blood Mononuclear Cells in Type 2 Diabetes: An Exploratory Study"

_medicina, 2022, doi:10.3390/medicina58121829_

Round 1

Reviewer 1 Report

In this paper entitled “Gene Expression Profiling of Peripheral Blood Mononuclear  Cells in Type 2 Diabetes: An Exploratory Study”  and submitted on Medicina (ISSN 1648-9144), Dr. Fakhoury and colleagues evaluated observationally the relative mRNA expression of macrophage phenotypic surface markers, cholesterol efflux proteins, scavenger receptors, and adenosine receptors in human circulating PBMCs isolated from patients with type 2 diabetes mellitus in order to phenotypically characterize and identify biomarkers for these ill-defined cells.

The work is interesting and well written in terms of both grammar and scientific writing, while the experimental design and statistical analyses (in exception of some inconsistences in terms of patients’ ages) are well performed. Several inaccuracies should be accurately revised in order to make the work suitable for publication. Please see below 

MAJOR

1. One the main limitations of the work is the lack of protein expression evaluation, as gene transcripts were evaluated, only. The protein levels of a gene quite often do not reflect the levels of the protein product. This limitation should be mentioned in the discussion. Whether being clinically relevant, authors should also consider in the future to evaluate the circulating levels of the investigated cytokines in patients’ sera

2. The general font stile of the entire work should be revised in order to uniformly it.  

3. Table 2 should be improved. The total number of subject should be reported in every group, while 18M, 12F annotations, (just as an example) should be well explained, i.e., Males (n=18) and Females (n=12). Moreover, “vs” annotations should be italicized. Please also revise all figure captions

4. P values should be included in the result section. “significantly” “no significant” should be accompanied by appropriate p values, even in case of non significance (i.e., p>0.05)

5. More references as a support of the statements in of the discussion should be included. For instance lines 265-268.

MINOR OBSERVATIONS

1. if possible in terms of length, I suggest mentioning al gene names in the abstract with their complete HUGO name. 

2. Lines 22-24 I suggest rephrasing the sentence as being the mRNAs of mentioned genes been studied by qPCR, not their protein product

3. Line 22 the function of metformin as type 2 diabetes medication should be mentioned in the abstract for non-expert readers

4. Lines 69-72 (or lines 275-279) This recently published review manuscript on A3 receptor characteristics and functions should be included https://pubmed.ncbi.nlm.nih.gov/34750517/ . Moreover, lines 72-73, I would mitigate this point as adenosine receptors are also currently employed as therapeutic targets in clinical trials for the treatment of various diseases

5. Line 78 “Although, “ the comma should be removed

6. Lines 101-119 the authors should check the font style

7. Line 146 melt curve details should be succinctly included

8. All figures a b c d panel annotations should be uniformed in terms of style

9. Line 241 “PBMC in circulation”  better “circulating PBMCs”

10. A possible relationship between A3 receptor and cholesterol has also been theorized doi: 10.3233/JAD-142223 and 

11. Liline 300 "ABCA1[23] “ please check the typo.

12. Lines 342-345 these sentences should be moved into the “conclusion” section

13. Lines 359-361Ethics approval code should be included

Reviewer 2 Report

The paper deals with profilisation of gene expression in PBMCs in Type 2 Diabetes confirmed individuals.

Searching for phenotypic markers produced by PBMCs was the aim of the work.

PBMCs is the frequently used sample type providing an interesting information about molecular processes based on immunological responses that are ongoing. Different diseases and stimulations correspond to the production of specific expressed molecules in PBMCs.

Thus, the work is actual and reliable when dealing with T2D as it is not the first work revealing interesting information about this disease based on profiling biomarkers in PBMCs.

REMARKS

1 I miss the general statement about the importance of PBMCs as a reliable sample for molecular research, pls, follow the down below upgrade in the 3rd paragraph of Introduction with proper citation.

“PBMCs is a sentinel sample, consisting of monocytes and lymphocytes that are sensitive to a variety of stimuli such as ongoing disease processes [https://doi.org/10.1002/pmic.202200026]. In early-onset of T2DM, PBMCs response is active as well [8]. In addition, an increase in pro-inflammatory cytokines and transcription factors in PBMC is associated with obesity and T2DM [9,10].”

2 In material and methods, provide a scheme figure of the sample preparation, analysis by PCR and bioinformatics assessment. This is due to fast understating for the reader what has been done in experimental part.

3 In conclusion part, provide authors future aims what could be else done in this area of investigation.

Round 2

Reviewer 1 Report

The ms has significantly been improved

Reviewer 2 Report

Authors have reacted to the given queries